# Surgical Management of Indeterminate Thyroid Nodules across Different World Regions: Results from a Retrospective Multicentric (the MAIN-NODE) Study

**DOI:** 10.3390/cancers15153996

**Published:** 2023-08-07

**Authors:** Gian Luigi Canu, Federico Cappellacci, Ahmed Abdallah, Islam Elzahaby, David Figueroa-Bohorquez, Eleonora Lori, Julie A. Miller, Sergio Zúñiga Pavia, Pilar Pinillos, Atcharaporn Pongtippan, Saleh Saleh Saleh, Salvatore Sorrenti, Chutintorn Sriphrapradang, Pietro Giorgio Calò, Fabio Medas

**Affiliations:** 1Department of Surgical Sciences, University of Cagliari, 09124 Cagliari, Italy; gianl.canu@unica.it (G.L.C.); fedcapp94@gmail.com (F.C.); pgcalo@unica.it (P.G.C.); 2Surgical Oncology, Mansoura University, Mansoura 35516, Egypt; drabdallah82@gmail.com (A.A.); islamabdouu2010@mans.edu.eg (I.E.); elbalkasaleh@mans.edu.eg (S.S.S.); 3Head and Neck Surgery, Hospital Universitario Nacional de Colombia, Bogotá 250247, Colombia; damfigueroabo@unal.edu.co (D.F.-B.); szunigap@unal.edu.co (S.Z.P.); pcpinillosn@unal.edu.co (P.P.); 4Department of Surgery, “Sapienza” University of Rome, 00185 Rome, Italy; eleonora.lori@uniroma1.it (E.L.); salvatore.sorrenti@uniroma1.it (S.S.); 5The Royal Melbourne Hospital and Epworth Hospital, Melbourne, VIC 3121, Australia; julie.miller@mh.org.au; 6Department of Pathology, Faculty of Medicine Ramathibodi Hospital, Mahidol University, Bangkok 10400, Thailand; kwang.atcharaporn@gmail.com; 7Department of Medicine, Faculty of Medicine Ramathibodi Hospital, Mahidol University, Bangkok 10400, Thailand; chutins@gmail.com

**Keywords:** indeterminate thyroid nodules, thyroid cancer, follicular thyroid neoplasm, thyroidectomy

## Abstract

**Simple Summary:**

Indeterminate thyroid nodules are characterized by a low-moderate risk of malignancy, which ranges from 5% to 40%. The management of these nodules is uneven among the current guidelines, with some being more prone to surgery, and others being more accommodating for conservative management. In this study, we aimed to evaluate the management of indeterminate thyroid nodules across different world regions. We found the Asian region to be more prone to active surveillance and Eastern regions to a more aggressive approach. Furthermore, current clinical practice seems to be diverging from current guidelines particularly regarding indications of fine-needle aspiration cytology. Our study underlines the need for homogenous guidelines and to discover new tools to assess the actual risk of malignancy in patients with indeterminate thyroid nodules.

**Abstract:**

Indeterminate thyroid nodules (ITNs) are characterized by an expected malignancy ranging from 5% to 30%, with most patients undergoing a diagnostic, rather than therapeutic, operation. The aim of our study was to compare the approach to ITNs across different regions of the world. In this retrospective, multicentric, international study, according to the WHO classification, we identified the South East Asian Region (SEAR), the Americas Region (AMR), the Eastern Mediterranean Region (EMR), the Europe Region (EUR), and the Western Pacific Region (WPR). One high-volume thyroid centre was included for each region. Demographic, preoperative, and pathologic data were compared among the different regions. Overall, 5737 patients from five high-volume thyroid centres were included in this study. We found that the proportion of ITNs over the global activity for thyroid disease was higher in the EUR (37.6%) than in the other regions (21.1–23.6%). In the EMR, the patients were significantly younger (with a mean of 43.1 years) than in the other regions (range, 48.8–57.4 years). The proportion of lobectomy was significantly higher in the WPR, where 83.2% (114/137) of patients received this treatment, than in the other regions, where lobectomies were performed in 44.1–58.1% of patients. The pathological diagnosis of malignancy was significantly higher in the SEAR centre, being over 60%, than in centres of the other regions, where it ranged from 26.3% to 41.3%. The occurrence of lymph node metastases was higher in the WPR (27.8%), AMR (26.9%), and EMR (20%) centres than in the EUR and SEAR centres, where it was lower than 10%. In summary, we found in our study different approaches and outcomes in the diagnosis and treatment of ITNs among countries. Overall, almost 60% of patients with ITNs who underwent surgery actually presented a benign disease, potentially undergoing an unnecessary operation.

## 1. Introduction

Thyroid carcinoma (TC) is the most common endocrine malignancy, affecting 0.2–1.5% of individuals worldwide, and its incidence has increased 300% over the past three decades [1]. TC is more common in women and affects a younger population than most malignancies, with a median age at diagnosis of 51 years. The rising incidence rate of TC is mostly related to the expanding use of high-quality imaging techniques, with an increase in the detection of thyroid nodules. Fine-needle aspiration cytology (FNAC) is the most accurate, rapid, safe, and cost-effective test for the evaluation of thyroid nodules, with a high specificity and sensitivity. Nevertheless, FNAC is particularly unreliable in differentiating between benign and malignant nodules that fall under the category of indeterminate thyroid nodules (ITNs) (class III and class IV according to Bethesda classification [2]). In fact, in these cases, the expected malignancy rates are 5–15% and 15–30%, respectively. Thus, most patients with ITNs undergo an operation that is potentially unnecessary, while representing a risk for surgical complications and a cost for health-care systems, even if we should consider that the adenoma–carcinoma sequence is gaining evidence. Furthermore, different managements of ITNs are possible, including—at two opposite sides—active surveillance and prompt surgery, varying widely from region to region, and also according to scientific society guidelines.

The aim of the MAIN-NODE (management of indeterminate thyroid nodules across different world regions) study was to compare the approach to ITNs across different regions of the world.

## 2. Materials and Methods

This was a retrospective, multicentric, international study. The study was previously registered at Clinicaltrials.gov (NCT05851404). The approval of the study was obtained from each centre’s local ethics committees. The study was conducted in accordance with the Declaration of Helsinki [3].

For the purpose of our study, according to the WHO classification, we identified the South East Asian Region (SEAR), the Americas Region (AMR), the Eastern Mediterranean Region (EMR), the Europe Region (EUR), and the Western Pacific Region (WPR). One high-volume thyroid centre was included for each region. We were unable to find a high-volume thyroid centre in the African Region. According to the 2020 European Society of Endocrine Surgeons (ESES) positional statement, a high-volume centre was defined if it performed at least 150 thyroidectomies per year [4].

The following characteristics were collected for each patient included in the study: demographics (sex and age), cytological classification according to the Bethesda System for Reporting Thyroid Cytopathology, the type of operation performed (total thyroidectomy vs. lobectomy, and central compartment lymph node dissection (CLND)), the pathological diagnosis (benign vs. malignant disease; in the case of thyroid carcinoma: histotype, size, extra thyroidal extension, lymph node metastasis, vascular invasion, and distant metastasis), and the risk of structural disease recurrence according to the ATA classification (low-, medium-, and high-risk).

Categorical variables were reported as cases (%); the differences among the groups were evaluated with a chi-squared test. Continuous variables were first tested for the normality of distribution, and then reported as the mean ± standard deviation; the differences among the groups were evaluated with an analysis of variance (ANOVA). The differences were considered as significant at *p* < 0.05. Calculations were made with MedCalc Vers 20.218.

## 3. Results

Overall, 5737 patients from five high-volume thyroid centres in five different WHO regions were included in this study (Table 1 and Figure 1). The operations for ITNs corresponded to 21.1% (253/1198) of all thyroidectomies performed in the SEAR centre, 22.4% (414/1845) in the AMR centre, 22.8% (283/1239) in the EMR centre, 37.6% (329/874) in the EUR centre, and 23.6% (137/581) in the WPR centre. The difference among the centres in the proportion of surgery for ITNs over the overall thyroidectomies was significant (*p* < 0.0001).

Patients with ITNs were included in the analysis; full results are reported in Table 2. A significant difference was observed among the centres regarding biological sex (*p* < 0.0001), with the male sex ranging from 8.5% (35/414) in the AMR centre to 28.9% (95/329) in the EUR centre. Overall, a significant difference was found regarding the age at operation, and ANOVA demonstrated significant differences among the SEAR centre (57.4 ± 14.1 years), the AMR centre (48.8 ± 14.2 years), the EMR centre (43.1 ± 13.8 years), the EUR centre (53.9 ± 14.5 years), and the WPR centre (52.7 ± 14.5 years); the only nonsignificant difference was observed between the EUR and WPR centres.

The cytological classification according to the Bethesda classification revealed a significant difference in the proportion of Bethesda III/Bethesda IV included in the study (*p* < 0.0001): Bethesda III was more frequent than Bethesda IV in the SEAR (95.3% (214/253)) and WPR (61.3% (84/137)) centres; conversely, Bethesda IV was more frequent than Bethesda III in the AMR (59.7% (247/414)), EMR (64% (181/283)), and EUR (67.5% (222/329)) centres.

Lobectomy was the preferred operation in the SEAR (58.5% (148/253)), AMR (53.9% (223/414)), and WPR (83.2% (114/137)) centres; on the other hand, total thyroidectomy was the most performed operation in the EMR (54.8% (155/283)) and EUR (55.9% (184/329)) centres. This difference was statistically significant (*p* < 0.0001). The dissection of the central compartment ranged from 1.5% (5/239) in the EUR centre to 25.6% (106/414) in the AMR centre, showing a significant difference among the centres (*p* < 0.0001).

The full report of pathological features is reported in Table 3. Pathological examination demonstrated a significant difference (*p* < 0.0001) in the malignancy rate among the centres, which was 60.1% (152/253) in the SEAR centre, 41.3% (171/414) in the AMR centre, 31.8% (90/283) in the EMR centre, 29.8% (98/329) in the EUR centre, and 26.3% (36/137) in the WPR centre.

The mean size of the tumour varied significantly among the centres, ranging from 22.8 ± 17.2 mm in the EUR centre to 31.2 ± 20.6 mm in the EMR centre. The occurrence of a microcarcinoma was significantly higher in the SEAR (28.9% (44/152)), AMR (37.4% (64/171)), and WPR (25% (9/36)) centres than in the EMR (14.4% (13/90)) and EUR (15.3% (15/98)) centres (*p* < 0.0001).

In addition, pathological examination revealed significant differences among the centres regarding the different histotypes. The classic variant of PTC was the most common histotype in SEAR (82.9% (126/152)), AMR (84.2% (144/171)), EMR (75.6% (68/90)), and WPR (63.9% (23/36)) centres, whereas follicular thyroid carcinoma was the most common histotype in the EUR (38.8% (38/98)). The occurrence of aggressive variants of PTC, including tall-cell and hobnail carcinoma, was higher in the SEAR (9.2% (14/152)) and EUR (14.4% (14/98)) centres than in the other centres (2.3% (4/171) in the AMR centre; 0 in the EMR centre; 2.8% (1/36) in the WPR centre).

The occurrence of extra thyroidal extension, lymph node metastasis, vascular invasion, and distant metastasis was significantly different among the centres. Particularly, the occurrence of extra thyroidal extension ranged from 5.3% (9/171) in the AMR centre to 16.7% (6/36) in the WPR centre, the occurrence of lymph node metastasis ranged from 8.2% (8/98) in the EUR centre to 26.9% (46/171) in the AMR centre, the occurrence of vascular invasion ranged from 7.6% (13/171) in the AMR centre to 25.0% (9/36) in the WPR centre, and distant metastases ranged from 0 cases in the EUR and WPR centres to 4.6% (7/152) in the SEAR centre.

Finally, a subanalysis of the risk of disease recurrence, according to the ATA classification, was performed on malignancies classified as differentiated thyroid cancers (Table 4). This analysis demonstrated a significant difference among the groups (*p* = 0.0417), with the occurrence of low-risk tumours ranging from 66.7% (24/36) in the WPR centre to 83.9% (73/87) in the EMR centre, the occurrence of intermediate-risk tumours ranging from 14.9% (13/87) in the EMR centre to 23.8% (38/160) in the AMR centre, and the occurrence of high-risk tumours ranging from 1.1% (1/87) in the EMR centre to 13.9% (5/36) in the WPR centre.

## 4. Discussion

The management of patients with ITNs at FNAC is demanding and problematic. Approximately 25% of all FNACs fall in this category, which represents a challenge for clinicians since malignancy, although relatively low (10–30%), cannot be safely excluded. Indeed, an increase in the occurrence of aggressive thyroid tumours has also been reported among these patients [5]. The management of ITNs includes mainly active surveillance and surgery; radiofrequency ablation is gaining evidence regarding its effectiveness in the treatment of ITNs but is still less employed in clinical practice [6]. Active surveillance is justified from the indolent behaviour of most thyroid cancers, which are usually slow-growing tumours. A recent study reported that over 70% of thyroid cancers have a tumour doubling time greater than 5 years; on the other hand, the same study revealed that approximately 10% of tumours present rapid-growing and aggressive behaviour [7].

In our study on surgical series, we evaluated the differences in the management of ITNs across the world, detecting significant differences among the different regions.

The proportion of ITNs over the global activity for thyroid disease was higher in the EUR (37.6%) than in the other regions (21.1–23.6%). This difference could represent a more frequent employment of FNAC in the management of thyroid nodules than in other regions.

In the EMR, the patients were significantly younger (with a mean age of 43.1 years) than in the other regions (range, 48.8–57.4 years), potentially reflecting a major diffusion of screening programs and a more aggressive management of thyroid nodules than in other countries.

In addition, in the SEAR, the occurrence of Bethesda III was 84.6%, exceedingly higher than in the other regions: if we exclude the WPR, where Bethesda III was the most common indication for surgery (61.3%), in the other regions, the most common cytological category was Bethesda IV, ranging from 59.7% to 67.5%. As also reported in the limits section, this difference could be explained by the different attitudes of pathologists in the classification of thyroid nodules.

The proportion of lobectomy was significantly higher in the WPR, where 83.2% (114/137) of patients received this treatment, than in the other regions, where lobectomies were performed in 44.1–58.1% of patients. In this regard, we should consider that in most cases, the indication for total thyroidectomy is not related to the indeterminate nodule itself, but usually reflects multinodular disease. The EUR centre, which had the highest occurrence of total thyroidectomies (55.9% (184/239)), is located in an iodine-deficient region, with a high occurrence of multinodular goitre and autoimmune thyroiditis. Thus, this difference among the countries could be explained by the different occurrence of thyroid pathologies rather than by the features of the ITNs.

The pathological diagnosis of malignancy was significantly higher in the SEAR centre, being over 60%, than in centres of the other regions, where it ranged from 26.3% to 41.3%.

The higher risk of malignancy (ROM) in the SEAR centre seems to be unrelated to a different prevalence of thyroid cancer in Asia than in other countries; in fact, according to data from GLOBOCAN 2018, thyroid carcinoma incidence was reported to be highest in North America, followed by Australia/New Zealand, eastern Asia, and Europe [8]. Despite these recent epidemiological data, our results, comparable to those of other studies, might be explained by the difference in the management of ITNs between Asian and Western countries [2,9,10,11,12,13,14,15,16].

Conservative management, represented by active surveillance without surgical intervention, promoted by the Japanese Thyroid Association (JTA) guidelines, is common in Asian practice for ITNs. JTA clinical guidelines recommend diagnostic surgery only for patients with ITNs with suspicious features, while those with benign clinical features and favourable ultrasonographic findings usually undergo active surveillance until suspicious features are detected. This more selective surgical approach results in a low resection rate, reducing unnecessary diagnostic surgery, and in a high ROM for surgically treated ITNs [9,10,11,12].

In contrast to Asian practice, in Western countries, in accordance with the recommendations of the 2015 American Thyroid Association (ATA) guidelines, active surveillance is not widely used in routine practice [13]. ATA guidelines recommend diagnostic surgery for all Bethesda IV nodules, but also for Bethesda III nodules with highly suspicious preoperative features. This management results in a higher resection rate and a lower ROM than in Asian countries. Although this approach prevents the missed diagnosis of cancer cases, it exposes patients to the possible complications of thyroidectomy [2,10,11,13].

Moreover, considering the economic impact of the two different approaches, especially in terms of cost effectiveness, active surveillance is a very low-cost procedure, requiring only blood tests and thyroid ultrasound, while the costs of diagnostic surgery are certainly higher, also considering the non-negligible costs related to surgical complications [14,15].

The occurrence of lymph node metastases was higher in the WPR (27.8%), AMR (26.9%), and EMR (20%) centres than in the EUR and SEAR centres, where it was lower than 10%. This finding seems to be related to the type of intervention performed; in fact, the occurrence of LNMs is proportional to the dissections of the central compartment performed.

Furthermore, it is interesting to observe that WPR had the highest occurrence of microcarcinomas but also of extrathyroidal extension and lymph node metastases. This might reflect an optimal selection of patients or a different behaviour of the thyroid neoplasms in this region, perhaps related to genetic or environmental factors.

In summary, we found in our study different approaches in the diagnosis and treatment of ITNs among countries. The risk of malignancy varied slightly among the centres included in our study, ranging from 26.3% (36/137) in the WPR to 60.1% (152/253) in the SEAR. Overall, approximately 40% of patients with ITNs who undergo surgery had thyroid cancer, and conversely, almost 60% of patients had benign disease.

In this scenario, a better stratification of the oncological risk seems to be pivotal. Recent advances in omics approaches (genomics, transcriptomics, proteomics, and metabolomics) are improving the understanding of molecular alterations associated with thyroid carcinoma initiation and progression, allowing for the identification of new biomarkers of malignancy that are useful for differentiating between benign and malignant nodules in cases of indeterminate cytology [17,18,19,20,21,22]. In the context of the omics analyses, a growing body of evidence is accumulating on the use of liquid biopsy for thyroid cancer, as already occurs for many other types of tumours (e.g., lung cancer). It represents a noninvasive approach that, using different technologies, analyses biomarkers released by cancer cells (e.g., circulating free nucleic acids, proteins, or metabolites) and detectable in body fluids (e.g., serum, saliva, or urine). The application of this approach in cases of ITN nodules may provide a more accurate preoperative estimation of the probability of malignancy, thus reducing unnecessary surgery for benign nodules [23,24,25,26].

Our study has several limitations. First, in our study, we included one centre for each region; thus, it is very likely that a centre is not representative of its own region. For this reason, the findings of our study are not generalizable. In addition, it is possible that a change in the selection of the patients occurred during the COVID-19 pandemic, period that was included in our study, leading to a different selection of patients, particularly during the hardest phase of the pandemic. Then, we should consider the differences in the interpretation of the FNAC by the pathologists, leading to a bias in our study. Particularly, the categorization in class III is the one most susceptible to the subjectivity and the experience of the pathologist.

## 5. Conclusions

The management of ITNs varies widely among countries, with Asian regions being more prone to active surveillance and Eastern regions to a more aggressive approach. Overall, almost 60% of patients with ITNs who underwent surgery actually presented a benign disease, potentially undergoing an unnecessary operation. Further studies on omics approaches to thyroid cancer are desirable and could provide clinicians with new tools in the future for the better selection of patients to be submitted to surgery.

## Figures and Tables

**Figure 1 cancers-15-03996-f001:**
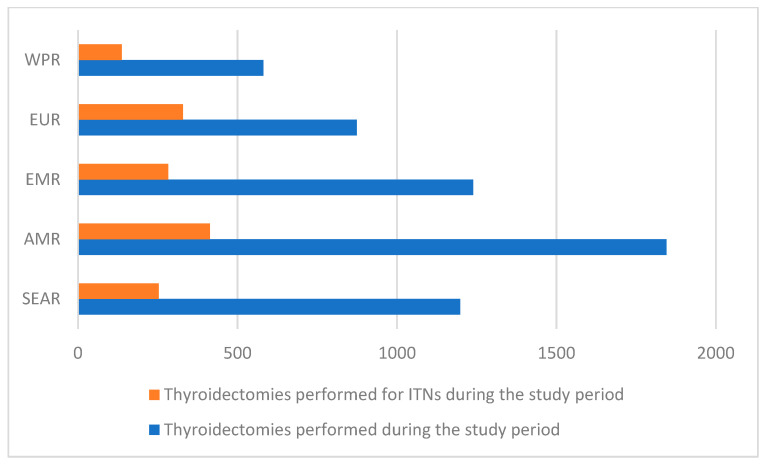
Thyroidectomies performed during the study period. ITNs: indeterminate thyroid nodules; WPR: Western Pacific Region; EUR: Europe Region; EMR: Eastern Mediterranean Region; AMR: The Americas Region; SEAR: South East Asian Region.

**Table 1 cancers-15-03996-t001:** Centres participating in the study across the WHO regions and number of surgeries performed during the study period (1 January 2019–31 December 2021).

WHO Regions	Centre	Thyroidectomies Performed during the Study Period *	Mean Thyroidectomies Performed per Year	Thyroidectomies Performed for ITNs during the Study Period **
South East Asian Region (SEAR)	Department of Medicine, Faculty of Medicine Ramathibodi Hospital, Mahidol University, Thailand	1198	399.3	253 (21.1%)
The Americas Region (AMR)	Head and Neck surgery, Hospital Universitario Nacional de Colombia, Bogotá D.C., Colombia	1845	615	414 (22.4%)
Eastern Mediterranean Region (EMR)	Surgical Oncology, Mansoura University, Mansoura, Egypt	1239	413	283 (22.8%)
Europe Region (EUR)	Department of Surgical Sciences, University of Cagliari, Italy	874	291.3	329 (37.6%)
Western Pacific Region (WPR)	The Royal Melbourne Hospital and Epworth Hospital	581	193.7	137 (23.6%)
Overall		5737	1912.3	1340 (23.4%)

* Thyroid surgeries for benign and malignant disease. ITNs: indeterminate thyroid nodules (class III and IV according to Bethesda reporting classification). ** The difference among the centres in the proportion of surgery for ITNs over the overall thyroidectomies was significant (*p* < 0.0001).

**Table 2 cancers-15-03996-t002:** Results of patients with indeterminate thyroid nodules included in the MAIN-NODE study.

	South East Asian Region	Americas Region	Eastern Mediterranean Region	Europe Region	Western Pacific Region	*p*
Patients with ITN	253	414	283	329	137	
Sex						<0.0001
Male	28 (11.1%)	35 (8.5%)	46 (16.3%)	95 (28.9%)	23 (16.8%)	
Female	225 (88.9%)	379 (91.5%)	237 (83.7%)	234 (71.1%)	114 (83.2%)	
Age (years)	57.4 ± 14.1	48.8 ± 14.2	43.1 ± 13.8	53.9 ± 14.5	52.7 ± 14.5	<0.0001 *
Preoperative FNAC						
Bethesda III	214 (84.6%)	167 (40.3%)	102 (36%)	107 (32.5%)	84 (61.3%)	<0.0001
Bethesda IV	39 (15.4%)	247 (59.7%)	181 (64%)	222 (67.5%)	53 (38.7%)	
Operation						
Total thyroidectomy	105 (41.5%)	191 (46.1%)	155 (54.8%)	184 (55.9%)	23 (16.8%)	<0.0001
Lobectomy	148 (58.5%)	223 (53.9%)	128 (45.2%)	145 (44.1%)	114 (83.2%)	
CLND	10 (4.0%)	106 (25.6%)	30 (10.6%)	5 (1.5%)	23 (16.8%)	<0.0001
Malignancies	152 (60.1%)	171 (41.3%)	90 (31.8%)	98 (29.8%)	36 (26.3%)	<0.0001

ITN: indeterminate thyroid nodules; FNAC: fine-needle aspiration cytology; CLND: central lymph node dissection. * ANOVA analysis revealed significant differences in Group 1 vs. Group 2 (*p* < 0.0001), Group 3 (*p* < 0.0001), Group 4 (*p* = 0.0359), and Group 5 (*p* = 0.0194); in Group 2 vs. Group 3 (*p* < 0.0001), Group 4 (*p* < 0.0001), and Group 5 (*p* < 0.0001); and in Group 3 vs. Group 4 (*p* < 0.0001) and Group 5 (*p* < 0.0001). The difference between Group 4 and Group 5 was not significant. Group 1 = South East Asian Region; Group 2 = Americas Region; Group 3 = Eastern Mediterranean Region; Group 4 = Europe Region; Group 5 = Western Pacific Region.

**Table 3 cancers-15-03996-t003:** Pathological examination of thyroid malignancies included in the MAIN-NODE study.

	South East Asian Region	Americas Region	Eastern Mediterranean Region	Europe Region	Western Pacific Region	*p*
Malignancies	152	171	90	98	36	
Size ≤ 10 mm	44 (28.9%)	64 (37.4%)	13 (14.4%)	15 (15.3%)	9 (25%)	<0.0001
Size (mm)	24.1 ± 19.2	16.4 ± 16.4	31.2 ± 20.6	22.8 ± 17.2	23.1 ± 20.9	<0.0001 *
Histotype						<0.0001
PTC, classic	126 (82.9%)	144 (84.2%)	68 (75.6%)	35 (35.7%)	23 (63.9%)	
PTC, aggressive	14 (9.2%)	4 (2.3%)	0	14 (14.4%)	1 (2.8%)	
FTC	3 (2.0%)	10 (5.8%)	13 (14.4%)	38 (38.8%)	9 (25.0%)	
HCC	4 (2.6%)	1 (0.6%)	5 (5.6%)	10 (10.2%)	3 (8.3%)	
Poorly differentiated	1 (0.7%)	1 (0.6%)	1 (1.1%)	0	0	
Anaplastic	1 (0.7%)	0	0	0	0	
Others	3 (2.0%)	11 (6.4%)	3 (3.3%)	1 (1.0%)	0	
Extra thyroidal extension	11 (7.2%)	9 (5.3%)	14 (15.6%)	16 (16.3%)	6 (16.7%)	0.0060
Lymph node metastasis	14 (9.2%)	46 (26.9%)	18 (20.0%)	8 (8.2%)	10 (27.8%)	0.0001
Vascular invasion	26 (17.1%)	13 (7.6%)	19 (21.1%)	12 (12.2%)	9 (25.0%)	0.0066
Distant metastasis	7 (4.6%)	1 (0.6%)	1 (1.1%)	0	0	0.0188

PTC: papillary thyroid carcinoma; FTC: follicular thyroid carcinoma; HCC: Hurtle cell thyroid carcinoma. * ANOVA analysis revealed significant differences in Group 1 vs. Group 2 (*p* < 0.0001) and Group 3 (*p* = 0.0001); in Group 2 vs. Group 3 (*p* < 0.0001), Group 4 (*p* < 0.0001), and Group 5 (*p* = 0.0022); and in Group 3 vs. Group 5 (*p* = 0.0002). The differences between Group 1 and Group 4 and Group 5, and between Group 4 and Group 5, were not significant. Group 1 = South East Asian Region; Group 2 = Americas Region; Group 3 = Eastern Mediterranean Region; Group 4 = Europe Region; Group 5 = Western Pacific Region.

**Table 4 cancers-15-03996-t004:** American Thyroid Association classification of risk of disease recurrence of differentiated thyroid cancers included in the MAIN-NODE study.

	South East Asian Region	Americas Region	Eastern Mediterranean Region	Europe Region	Western Pacific Region	*p*
DTC	148	160	87	97	36	0.0417
Low risk	110 (74.3%)	108 (67.5%)	73 (83.9%)	66 (68%)	24 (66.7%)	
Intermediate risk	29 (19.6%)	38 (23.8%)	13 (14.9%)	18 (18.6%)	7 (19.4%)	
High risk	9 (6.1%)	14 (8.8%)	1 (1.1%)	13 (13.4%)	5 (13.9%)	

DTC: differentiated thyroid cancer.

## Data Availability

Anonymized data will be made available after reasonable request to fabiomedas@unica.it.

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
