# Peer review of "Surgical Management of Indeterminate Thyroid Nodules across Different World Regions: Results from a Retrospective Multicentric (the MAIN-NODE) Study"

_cancers, 2023, doi:10.3390/cancers15153996_

Round 1
Reviewer 1 Report
This is a very well written article.
I have only minor points.
Please explain what „high- volume thyroid centre” means.
Table 1 could be better displayed as a figure (bar graph)
Reviewer 2 Report
The manuscript entitled "Management of indeterminate thyroid nodules across different world regions: results from a retrospective multicentric (the MAIN-NODE) study" is a retrospective multicenter study focused on the management of indeterminate thyroid nodules across different Countries and World Regions.
The Authors claims that they have analyzed the ways different Countries are managing the ITNs. However, they have only considered the “surgical” approach (i.e., the number of thyroidectomies performed in patients with ITNs). There is no mention regarding any ancillary methods used to increase the accuracy of FNA cytology, that have possibly been used to select patients for surgery. It cannot be ignored that between the two approaches indicated by the Authors and consisting in “active surveillance”, with repeated sonographic evaluation and repeated FNA cytology, and “prompt surgery” with lobectomy or total thyroidectomy, there is a wide spectrum of possible methods aimed to increase the accuracy of FNA cytology in these group of lesions, as suggested also by the ATA (2015 American Thyroid Association Management Guidelines for Adult Patients with Thyroid Nodules and Differentiated Thyroid Cancer, Thyroid. Jan 2016, 26(1): 1-133) as well by the many different guidelines reported by several International and National Thyroid/Endocrine Societies. In particular, in the USA many different test-methods are currently commercially available to obtain a molecular profile of thyrocytes obtained at FNA cytology. In addition, the use of protein markers, such as Galectin-3 and HMBE-1, have been proposed and validated in retrospective (Lancet. 2001;357:1644–50) as well as in prospective multicentric studies (Lancet Oncol. 2008;9:543-9) and are routinely used in many Countries to increase cancer detection capacity among cytologically ITNs. Finally, the use of FDG-PET has been proposed for the same purpose. All these test-methods have been introduced to increase accuracy of preoperative FNA cytology and to fill the gap between the two already mentioned approaches of simple “active surveillance” and “prompt surgery”. A comparative analysis among all these ancillary test-methods, has been published in 2017, where the diagnostic performance, feasibility and the cost of the different test-methods, used to address the question whether to indicate surgery or not, were analyzed in different studies in many Countries (Oncotarget, 2017;8:49421-49442). More recently, other new methods have been proposed, including those mentioned by the Authors and consisting in approaches based on proteomic analysis and on liquid biopsy.
Major points
1. The study has major limitations. It reports data of ITNs that were surgically treated, without reporting those that were not subjected to surgery. The Authors, in fact, have collected and analyzed data concerning only the “surgical” approach, without considering the total number of ITNs and the outcome of those that were not operated and were directed to active monitoring. Moreover, there is no indication whether any ancillary method have been used to better stratify patients with indeterminate thyroid nodules. In this regard, the title is inappropriate, since it should refer more to a “surgical” management of indeterminate thyroid nodules. There is no mention, in fact, regarding the ITNs that were not treated by surgery and how they were managed. It appears clear that the surgical selection, performed in the different Centers, resulted in different final histological outcomes. In the SEAR group there is a high incidence of cancer among ITNs subjected to surgery (152/253, with a rate of malignancy equal to 60%). Conversely, a much lower malignancy rate at surgery was observed in the WPR (36/137, equal to 26.3%). In the other Centers included in the study, the rate of malignancy detected at surgery ranged from 29,8%, observed in ER, to 41%, observed in AMR. In my opinion, these data are relevant for the manuscript and should deserve a separate Table and a dedicate paragraph too. This suggests that the surgical decision in SEAR was based on better selection criteria, compared to other Centers and, most notably, compared to those applied in WPR. It would be interesting to know in what they differ (a better sonographic evaluation? any additional tests?). Is this type of selection responsible for any missing diagnosis of cancer that could have been recognized using a more aggressive diagnostic approach? Once again, the close follow-up of the ITNs that were not histologically verified is essential to answer this question.
2. The study doesn’t add any particular new data regarding what is already known on this topic. In the introduction, the Authors stated that “indeterminate thyroid nodules are characterized by a low-moderate risk of malignancy, which ranges from 5% to 40% and in the conclusion, it is stated that “almost 60% of patients with ITNs who underwent surgery actually presented a benign disease, potentially undergoing an unnecessary operation”. There is no indication on how to solve the problem and how the different Countries have managed the diagnostic and therapeutic approach. The issue is still open and it can be solved, as also state by the Authors, by improving the preoperative recognition of malignancy, using the most accurate and possibly the less expensive methods available.
3. The study has focused the attention only on the pathological examination of thyroid malignancies. No data have been reported regarding the histological description of the ITNs that were diagnosed as benign, which represent in almost all Centers (excluding the SEARS one) the vast majority. In order to verify the cytological diagnosis, it would be useful to know, at least, how many of them were true neoplastic lesions.
Minor points
1. the legend of Table 2 is not clearly indicated. The Groups indicated as Group 1, 2, 3, 4, 5 are probably the same that are indicated in the same legend as SEAR, WPR, AMR, EMR, EUR.
2. The Authors have mentioned the THYCOVID study in the reference list (Ref. n. 5). It is a study published by one of the Authors in the Journal “The Lancet” in 2023. The THYCOVID study was mentioned in Table 2, Table 3 and Table 4. However, in no part of the manuscript is explained whether the results reported in the Tables refer to data obtained from this previously published study. Conversely, the present manuscript, that is submitted for the publication in the Journal “Cancers”, apparently is based on data retrospectively obtained from patients recruited in another trial, with another acronymous, namely the MAIN-NODE and with a distinct registration number (NCT05851404) at the clinicaltrials.gov web site. The Authors should explain whether they have used the same data set of the THYCOVID study or not.
Author Response
The manuscript entitled "Management of indeterminate thyroid nodules across different world regions: results from a retrospective multicentric (the MAIN-NODE) study" is a retrospective multicenter study focused on the management of indeterminate thyroid nodules across different Countries and World Regions.
The Authors claims that they have analyzed the ways different Countries are managing the ITNs. However, they have only considered the “surgical” approach (i.e., the number of thyroidectomies performed in patients with ITNs). There is no mention regarding any ancillary methods used to increase the accuracy of FNA cytology, that have possibly been used to select patients for surgery. It cannot be ignored that between the two approaches indicated by the Authors and consisting in “active surveillance”, with repeated sonographic evaluation and repeated FNA cytology, and “prompt surgery” with lobectomy or total thyroidectomy, there is a wide spectrum of possible methods aimed to increase the accuracy of FNA cytology in these group of lesions, as suggested also by the ATA (2015 American Thyroid Association Management Guidelines for Adult Patients with Thyroid Nodules and Differentiated Thyroid Cancer, Thyroid. Jan 2016, 26(1): 1-133) as well by the many different guidelines reported by several International and National Thyroid/Endocrine Societies. In particular, in the USA many different test-methods are currently commercially available to obtain a molecular profile of thyrocytes obtained at FNA cytology. In addition, the use of protein markers, such as Galectin-3 and HMBE-1, have been proposed and validated in retrospective (Lancet. 2001;357:1644–50) as well as in prospective multicentric studies (Lancet Oncol. 2008;9:543-9) and are routinely used in many Countries to increase cancer detection capacity among cytologically ITNs. Finally, the use of FDG-PET has been proposed for the same purpose. All these test-methods have been introduced to increase accuracy of preoperative FNA cytology and to fill the gap between the two already mentioned approaches of simple “active surveillance” and “prompt surgery”. A comparative analysis among all these ancillary test-methods, has been published in 2017, where the diagnostic performance, feasibility and the cost of the different test-methods, used to address the question whether to indicate surgery or not, were analyzed in different studies in many Countries (Oncotarget, 2017;8:49421-49442). More recently, other new methods have been proposed, including those mentioned by the Authors and consisting in approaches based on proteomic analysis and on liquid biopsy.
Thank you very much for these observations and for the reference to the articles of Bortolazzi et al.
Major points
- The study has major limitations. It reports data of ITNs that were surgically treated, without reporting those that were not subjected to surgery. The Authors, in fact, have collected and analyzed data concerning only the “surgical” approach, without considering the total number of ITNs and the outcome of those that were not operated and were directed to active monitoring.
According to your suggestions, we have modified the title to "Surgical management of indeterminate thyroid nodules across different world regions: results from a retrospective multicentric (the MAIN-NODE) study". In fact, our series derives from surgical centers. We have no data regarding active surveillance.
- Moreover, there is no indication whether any ancillary method have been used to better stratify patients with indeterminate thyroid nodules. In this regard, the title is inappropriate, since it should refer more to a “surgical” management of indeterminate thyroid nodules. There is no mention, in fact, regarding the ITNs that were not treated by surgery and how they were managed. It appears clear that the surgical selection, performed in the different Centers, resulted in different final histological outcomes. In the SEAR group there is a high incidence of cancer among ITNs subjected to surgery (152/253, with a rate of malignancy equal to 60%). Conversely, a much lower malignancy rate at surgery was observed in the WPR (36/137, equal to 26.3%). In the other Centers included in the study, the rate of malignancy detected at surgery ranged from 29,8%, observed in ER, to 41%, observed in AMR. In my opinion, these data are relevant for the manuscript and should deserve a separate Table and a dedicate paragraph too. This suggests that the surgical decision in SEAR was based on better selection criteria, compared to other Centers and, most notably, compared to those applied in WPR. It would be interesting to know in what they differ (a better sonographic evaluation? any additional tests?). Is this type of selection responsible for any missing diagnosis of cancer that could have been recognized using a more aggressive diagnostic approach? Once again, the close follow-up of the ITNs that were not histologically verified is essential to answer this question.
Thank you very much for these observations. We discussed this issue in the Discussion section; particularly, we osberved that the higher risk of malignancy (ROM) in the SEAR centre seems to be unrelated to a different prevalence of thyroid cancer in Asia than in other countries. Conservative management is common in Asian practice for ITNs. JTA clinical guidelines recommend diagnostic surgery only for patients with ITNs with suspicious features, while those with benign clinical features and favourable ultrasonographic findings usually undergo active surveillance until suspicious features are detected. This more selective surgical approach results in a low resection rate, reducing unnecessary diagnostic surgery, and in a high ROM for surgically treated ITNs.
Unfortunately, none of the included center was able to employ molecular diagnostics on FNAC.
- The study doesn’t add any particular new data regarding what is already known on this topic. In the introduction, the Authors stated that “indeterminate thyroid nodules are characterized by a low-moderate risk of malignancy, which ranges from 5% to 40% and in the conclusion, it is stated that “almost 60% of patients with ITNs who underwent surgery actually presented a benign disease, potentially undergoing an unnecessary operation”. There is no indication on how to solve the problem and how the different Countries have managed the diagnostic and therapeutic approach. The issue is still open and it can be solved, as also state by the Authors, by improving the preoperative recognition of malignancy, using the most accurate and possibly the less expensive methods available.
Thank you for your observation. The aim of this paper was to describe the current situation in different regions regarding surgical treatment of ITNs. As you recognized in the second sentence, we suggested in the discussion that molecular techniques should be exploited in the diagnostic of ITNs to stratify the risk of malignancy. - The study has focused the attention only on the pathological examination of thyroid malignancies. No data have been reported regarding the histological description of the ITNs that were diagnosed as benign, which represent in almost all Centers (excluding the SEARS one) the vast majority. In order to verify the cytological diagnosis, it would be useful to know, at least, how many of them were true neoplastic lesions.
Unfortunately, we did not collect data on benign lesions. If you feel this could be relevant to the work, we could retrieve this data. Usually, among benign lesions, we observe 40%-45% of follicular adenomas and 55%-60% of multinodular goiter.
Minor points
- the legend of Table 2 is not clearly indicated. The Groups indicated as Group 1, 2, 3, 4, 5 are probably the same that are indicated in the same legend as SEAR, WPR, AMR, EMR, EUR.
We have clarified the legend. - The Authors have mentioned the THYCOVID study in the reference list (Ref. n. 5). It is a study published by one of the Authors in the Journal “The Lancet” in 2023. The THYCOVID study was mentioned in Table 2, Table 3 and Table 4. However, in no part of the manuscript is explained whether the results reported in the Tables refer to data obtained from this previously published study. Conversely, the present manuscript, that is submitted for the publication in the Journal “Cancers”, apparently is based on data retrospectively obtained from patients recruited in another trial, with another acronymous, namely the MAIN-NODE and with a distinct registration number (NCT05851404) at the clinicaltrials.gov web site. The Authors should explain whether they have used the same data set of the THYCOVID study or not.
We are very sorry for this mistake. This was a clerical error. We just used the template of our previous work. The studies are focused on indeterminate thyroid nodules, but are derived from different series, and, concurrently, are registered as different works.
Reviewer 3 Report
Thank you very much for possibility of reviewing this paper.
The aim of the study was to analyze the management of indeterminate thyroid nodules across different world regions. The clinical value of the study is incredible high, because the undertaken topic is still unresolved. So, it seems, that the knowledge how clinicians in some other countries treat the patients with these thyroid nodules is very important. There are some minor points, which should be reanalyzed:
# In the line 60 when you introduced the abbreviation of indeterminate thyroid nodules (ITNs) consequently you should use it in the next text, not the whole name again like for example in the line 62 or 183
# It would be great to know some more details of each high volume centers included to the study like their profiles, number of all operations per year, number of thyroid surgeries per year, some emergency procedures, number of thyroidectomies due to thyroid cancer, etc.,
# The date of the pandemic outbreak is March 11, 2020, so do not you think this particular time may have changed some crucial data during the study period
# in the line 100 the authors wrote the abbreviation AR instead of I think AMR, so correct,
# It would be nice to know how many AUS and FLUS diagnoses were in category III Bethesda classification in each region and what histopathology they obtained from them, especially from AUS
# It would be great to know what histotypes were obtained from ITNs type III and IV separately
# If the occurrence of aggressive variants of PTC, including tall cell and hobnail carcinoma, was higher in the SEAR (9.2% [14/152]), why this center does not decide to perform more radical surgeries (not predominantly lobectomy) what we can observe, for example, in EUR center (14.4% [14/98])
# The same question is regarding distant metastases, the highest percent we observe in SEAR center or the number of malignancy in ITNs (60%)
# how is it possible that in the SEAR center the percent of distant metastases or aggressive carcinomas is almost the highest, but the percent of high risk of recurrence is almost the lowest
# In the line 215 the abbreviation ER should be changed for EUR previously and officially introduced and used in the text of manuscript
# the same situation in the line 251
Thank you so much.
Author Response
Thank you very much for possibility of reviewing this paper.
The aim of the study was to analyze the management of indeterminate thyroid nodules across different world regions. The clinical value of the study is incredible high, because the undertaken topic is still unresolved. So, it seems, that the knowledge how clinicians in some other countries treat the patients with these thyroid nodules is very important. There are some minor points, which should be reanalyzed:
# In the line 60 when you introduced the abbreviation of indeterminate thyroid nodules (ITNs) consequently you should use it in the next text, not the whole name again like for example in the line 62 or 183
The abbreviation has been substitued throughout the manuscript.
# It would be great to know some more details of each high volume centers included to the study like their profiles, number of all operations per year, number of thyroid surgeries per year, some emergency procedures, number of thyroidectomies due to thyroid cancer, etc.,
In table 1 we have added the yearly number of operations performed by each center.
# The date of the pandemic outbreak is March 11, 2020, so do not you think this particular time may have changed some crucial data during the study period
Yes, we have added this issue in the limitation of the study.
# in the line 100 the authors wrote the abbreviation AR instead of I think AMR, so correct,
Done, thank you
# It would be nice to know how many AUS and FLUS diagnoses were in category III Bethesda classification in each region and what histopathology they obtained from them, especially from AUS
# It would be great to know what histotypes were obtained from ITNs type III and IV separately
Unfortunately, we have only aggregated data for Bethesda III and IV and we cannot retrieve these informations.
# If the occurrence of aggressive variants of PTC, including tall cell and hobnail carcinoma, was higher in the SEAR (9.2% [14/152]), why this center does not decide to perform more radical surgeries (not predominantly lobectomy) what we can observe, for example, in EUR center (14.4% [14/98])
This is an extremely relevant observation. Unfortunately, the diagnosis of aggressive thyroid cancer (including tall cell and hobnail variant) is a pathological and postoperative diagnosis. We underlined at the end of the discussion the need to improve the preoperative risk assessment, including the development of multi-omic approach to select patients with aggressive cancers requiring aggressive surgery.
# The same question is regarding distant metastases, the highest percent we observe in SEAR center or the number of malignancy in ITNs (60%)
In this case, if distant metastases are already present, the indication for a CLND is controversial; in fact, we are treating cancers that have already distant metastases, and curative surgery is not feasible: a CLND could increase postoperative complications without significant advantages in terms of survival.
# how is it possible that in the SEAR center the percent of distant metastases or aggressive carcinomas is almost the highest, but the percent of high risk of recurrence is almost the lowest
This could be explained by the fact that we only included in the ATA stratification for risk of disease recurrence (Table 4) only patients with differentiated thyroid carcinoma (DTC), according to the ATA guidelines. Patients with distant metastases were patients with anaplastic, poorly differentiated, and medullary thyroid carcinomas, thus these patients were not stratified for risk of recurrence, according to ATA guidelines.
# In the line 215 the abbreviation ER should be changed for EUR previously and officially introduced and used in the text of manuscript
# the same situation in the line 251
Done, thank you
Thank you so much.
Thank you very much for your suggestions.